# Intercalated PtCo Electrocatalyst of Vanadium Metal Oxide Increases Charge Density to Facilitate Hydrogen Evolution

**DOI:** 10.3390/molecules29071518

**Published:** 2024-03-28

**Authors:** Jingjing Zhang, Wei Deng, Yun Weng, Jingxian Jiang, Haifang Mao, Wenqian Zhang, Tiandong Lu, Dewu Long, Fei Jiang

**Affiliations:** 1School of Chemical and Environmental Engineering, Shanghai Institute of Technology, Shanghai 201418, China; zhangjingjingsit@126.com (J.Z.); jjx@sit.edu.cn (J.J.); maohaifang2000@126.com (H.M.); zhangwq2004@163.com (W.Z.); lutiandong2003@163.com (T.L.); 2State Key Laboratory for Modification of Chemical Fibers and Polymer Materials, College of Textile, Donghua University, Shanghai 201620, China; wendy_weng@163.com; 3Key Laboratory in Interfacial Physics and Technology, Shanghai Institute of Applied Physics, Chinese Academy of Sciences, Shanghai 201800, China; longdewu2010@126.com

**Keywords:** hydrogen evolution reaction, electrocatalysts, polymetallic compounds, hydrogen metal oxide bronzes

## Abstract

Efforts to develop high-performance electrocatalysts for the hydrogen evolution reaction (HER) are of utmost importance in ensuring sustainable hydrogen production. The controllable fabrication of inexpensive, durable, and high-efficient HER catalysts still remains a great challenge. Herein, we introduce a universal strategy aiming to achieve rapid synthesis of highly active hydrogen evolution catalysts using a controllable hydrogen insertion method and solvothermal process. Hydrogen vanadium bronze H_x_V_2_O_5_ was obtained through controlling the ethanol reaction rate in the oxidization process of hydrogen peroxide. Subsequently, the intermetallic PtCoVO supported on two-dimensional graphitic carbon nitride (g-C_3_N_4_) nanosheets was prepared by a solvothermal method at the oil/water interface. In terms of HER performance, PtCoVO/g-C_3_N_4_ demonstrates superior characteristics compared to PtCo/g-C_3_N_4_ and PtCoV/g-C_3_N_4_. This superiority can be attributed to the notable influence of oxygen vacancies in H_x_V_2_O_5_ on the electrical properties of the catalyst. By adjusting the relative proportions of metal atoms in the PtCoVO/g-C_3_N_4_ nanomaterials, the PtCoVO/g-C_3_N_4_ nanocomposites show significant HER overpotential of η_10_ = 92 mV, a Tafel slope of 65.21 mV dec^−1^, and outstanding stability (a continuous test lasting 48 h). The nanoarchitecture of a g-C_3_N_4_-supported PtCoVO nanoalloy catalyst exhibits exceptional resistance to nanoparticle migration and corrosion, owing to the strong interaction between the metal nanoparticles and the g-C_3_N_4_ support. Pt, Co, and V simultaneous doping has been shown by Density Functional Theory (DFT) calculations to enhance the density of states (DOS) at the Fermi level. This augmentation leads to a higher charge density and a reduction in the adsorption energy of intermediates.

## 1. Introduction

To resolve the global energy crisis, the desirable search for renewable energy sources to substitute traditional fossil energy has attracted wide attention all over the world, including metal–air batteries, fuel cells, and electrochemical water decomposition [1,2]. The clean, renewable, and abundant nature of hydrogen energy positions it as one of the most promising energy carriers [3,4]. Over the past dozen years, electrocatalysts including oxides and alloys [5,6], phosphide [7,8], nitrides [9,10], selenides [11], and carbides [12,13] have shown remarkable HER activity. However, high-efficient, low-cost electrochemical water splitting development still faces great challenges.

Traditional hydrogen production strategies cannot avoid the emission of carbon dioxide, while electrically driven water splitting technology is a promising technology for hydrogen production [14,15,16]. In the process of water electrolysis, concerning how to obtain high-energy density and pure hydrogen effectively and reduce the overpotential, electrocatalysts play a crucial role. Over the past few decades, platinum (Pt)-based noble metals have been considered benchmark HER electrocatalysts which exhibit extremely low overpotential [17]. In spite of the positive features of Pt-based electrocatalysts, the great number of inconveniences (hard to synthesize) and shortcomings (low abundance and high price) still impede their practical applications [18]. Consequently, it is particularly desirable to seek high-efficient and durable catalysts for water electrolysis to reduce noble Pt content and overpotential, thus improving the energy conversion efficiency [19].

In this study, we present a novel method utilizing hydrogen insertion vanadium powder (V) to manipulate the reaction rate of hydrogen peroxide (H_2_O_2_) in the presence of ethanol. By controlling the reaction rate through the presence of ethanol, we can successfully obtain hydrogen metal oxide bronzes (HVOs), such as hydrogen vanadium bronze. Moreover, the HVOs exhibit excellent dispersion and stability in solvents devoid of water, specifically alcoholic-based solvents. This characteristic is highly advantageous for enhancing device stability and facilitating processing. Previous reports have shown that coupling metal nanoparticles with carbon substrates can effectively improve electrocatalytic performance and enhance stability due to the effective surface corrosion inhibition and excellent electron transport properties of carbon materials [20]. Therefore, g-C_3_N_4_ is chosen as the substrate in this study. Subsequently, the intermetallic PtCoVO anchored on two-dimensional g-C_3_N_4_ nanosheets is prepared by a one-pot solvothermal process at the oil/water interface. The PtCoVO/g-C_3_N_4_ catalyst, prepared in the aforementioned manner, demonstrates exceptional performance in the HER, which can be attributed to the substantial impact of oxygen vacancies present in HVOs on the electrical properties of the catalyst, resulting in enhanced HER performance [21]. As anticipated, the PtCoVO/g-C_3_N_4_ catalyst demonstrates excellent performance in an alkaline HER, showcasing a low overpotential of η_10_ = 92 mV and a small Tafel slope of 65.21 mV dec^−1^. Additionally, the catalyst exhibits remarkable stability, comparable to the near state-of-the-art Pt/C 20% catalyst (90.6 mV, 10 mA cm^−2^). Detailed structural investigations reveal that the hydrogen insertion strategy modulates the effect of oxygen content in HVOs on electrical properties, which might result in a special V active center that enhances HER activity. 

## 2. Results and Discussion

### 2.1. Morphologies and Structures

The tunable polymetallic nanocomposites PtCoVO/g-C_3_N_4_ were successfully synthesized through a solvothermal method, as illustrated in Figure 1, which utilized three kinds of metal precursors, cobalt acetylacetonate (Co(acac)_3_), H_2_PtCl_6_·6H_2_O, and vanadium bronze (here, we use H_x_V_2_O_5 to_ represent HVOs), grown on two-dimensional g-C_3_N_4_ nanosheets. In particular, the unique low-temperature solution-processed hydrogen vanadium bronze H_x_V_2_O_5_ was synthesized through a regulable hydrogen insertion method. Under ambient temperature conditions, vanadium powder was gradually oxidized by H_2_O_2_ in the existence of ethanol. Through the carefully controlled, sluggish reaction rate facilitated by ethanol, the solution-processed hydrogen metal oxide bronze H_x_V_2_O_5_ was successfully obtained [22]. The H_x_V_2_O_5_ color changed from yellow to brown during the preparation process, as can be seen in Appendix A. Subsequently, the polymetallic PtCoVO supported on two-dimensional g-C_3_N_4_ nanosheets was synthesized by using a one-pot solvothermal method at the oil/water interface. The enlarged PtCoVO surface structure was selected as the computational model for the illustration of as-prepared PtCoVO/g-C_3_N_4_ electrocatalysts. For the purpose of comparison, our focus was directed toward three distinct materials with varying chemical compositions that were prepared in advance: PtCo/g-C_3_N_4_, PtCoV/g-C_3_N_4_, and PtCoVO/g-C_3_N_4_, respectively. We utilized V powder to directly prepare PtCoV/g-C_3_N_4_ to compare the different influences between V powder and H_x_V_2_O_5_ bronzes. (Here, we denote PtCo/g-C_3_N_4_ as prepared by adding no vanadium, PtCoV/g-C_3_N_4_ as prepared by adding V powder, and PtCoVO/g-C_3_N_4_ as prepared by H_x_V_2_O_5_) By adjusting the metal atoms, Pt, Co, and V, to appropriate molar ratio, the composite materials were meticulously optimized to achieve desired tunability in the elemental components. A detailed description of the synthesis process for composites with varying proportions is provided in the experimental section.

### 2.2. Electrocatalytic Enhancement Mechanism

In order to thoroughly investigate the mechanism behind the hydrogen evolution reaction (HER) enhancement and the improved properties of PtCoVO/g-C_3_N_4_, the crystal structures of PtCo/g-C_3_N_4_ and PtCoVO/g-C_3_N_4_ (illustration of Figure 2a) were constructed [23]. Owing to having the identical supporter on two-dimensional g-C_3_N_4_ nanosheets, we focused on PtCo and PtCoVO for simplified models. DFT studies were executed to survey the effect of PtCo and PtCoVO on the atomic-level electronic structure [24]. A potential electrocatalyst is characterized by favorable hydrogen adsorption free energy (ΔG_H*_), where a ΔG_H*_ value close to zero can lead to enhanced hydrogen evolution activity due to an optimal equilibrium between the absorption and desorption of hydrogen atoms from the active site. The PtCo free energy change (ΔG_H*_) in the hydrogen adsorption value deviates significantly from the optimal value (≈0 eV) [25], that is, the PtCo surface is inactive in Figure 2a. However, the ΔG_H*_ value of PtCoVO combined with an H atom with HMOs (hydrogen metal oxide bronzes) is much smaller, 0.18 eV, which is conducive to the hydrogen evolution reaction [26]. The structure diagram of PtCo (H*) and PtCoVO (H*) is shown in Appendix A. Likewise, the density of states (DOS) analysis demonstrates that the PtCoVO catalyst exhibits a significantly higher density of states in close proximity to the Fermi level compared to the other substrates [27], when compared with PtCo/g-C_3_N_4_ (Figure 2b,c). When H_x_V_2_O_5_ (HMOs) is added to frame, the DOS of PtCoVO is much higher than that PtCo, which indicates that the implanting of H_x_V_2_O_5_ significantly enhances the DOS. This indicates that V is the active site in PtCoVO. The high DOS indicates increased electron transfer, thereby accelerating the electrocatalytic reaction and resulting in a significant improvement in HER performance [28]. Appendix A also verify this conclusion. Additionally, the intensity of the DOS is strongly correlated with the conductivity [29]. The enhanced charge transfer dynamics of PtCoVO can be attributed to its higher DOS intensity, which aligns with the reduced charge transfer resistance. In addition, we can derive data from Appendix A showing that the DOS of V in PtCoVO at the Fermi level is 2.511 eV, which is much higher than the DOS of PtCo, but close to that of PtCoVO. Therefore, we can ascertain that the introduction of V improves the DOS. The catalytic activity is closely related to the catalyst surface free energy of the hydrogen adsorption [30]. Moreover, the charge density difference model of PtCo and PtCoVO is depicted in Figure 2d,e. The charge density distribution of PtCo and PtCoVO indicate that doping with V atoms exacerbates the inhomogeneity of the distribution of electric charge. The HER activity of the catalyst is thus increased, so doping with V can introduce a large number of active catalytic sites, which has a crucial advantage for the successful release of hydrogen [31,32]. In addition, we also plot the molecular models of PtCoV and PtCoV (*), the band structure, and the PDOS (partial density of states), which can be shown in Appendix A.

Figure 3a illustrates the scanning electron microscopy (SEM) image of the as-synthesized trimetallic PtCoVO nanoparticles decorated onto the g-C_3_N_4_ nanosheets. The corresponding energy dispersive X-ray (EDX) spectrum confirmed the uniform distribution of Co, Pt, V, C, N, and O atoms throughout the material (Appendix A). It appeared that Pt, Co, V, C, N, and O existed in PtCoVO/g-C_3_N_4_. Weight (Wt) ratios of PtCoVO/g-C_3_N_4_ elements were measured by EDX analysis. The Wt ratio of each element in the PtCoVO/g-C_3_N_4_ was 26.80:13.86:21.50:9.08:9.56:20.20 (C:N:O:Pt:Co:V). The precise mass ratio of each element in PtCoVO/g-C_3_N_4_ was determined using inductively coupled plasma emission spectrometry (ICP-AES), which was 6.43:7.02:15.95 (Pt:Co:V), consistent with the mass ratio data measured by EDX (Appendix A). The incorporation of the PtCoV nanoalloy into the g-C_3_N_4_ nanosheets was further confirmed by the high-resolution transmission electron microscopy (HRTEM) image shown in Figure 3b, which exhibits a lattice fringe spacing of 0.226 nm. This lattice was identified as the (111) crystallographic plane of Pt, according to the Powder Diffraction File (PDF#04-0802) [33]. Meanwhile, structural information could be extracted by analyzing the fast Fourier transform (FFT) and inverse fast Fourier transform (IFFT) modes of the selected region for details of the structure [34] (Figure 3c,d). The lattice stripes observed in Figure 3e, Selected Area Electron Diffraction (SAED) images, indicate that they originated from the (111) and (200) crystallographic planes of Pt, as identified by the (PDF#04-0802). The FFT images shown in Figure 3f provide additional confirmation of the lattice spacing of approximately 2.265 nm (equivalent to 10 lattice distances) between the atomic layers of the exposed (111) facet. The use of High Angle Annular Dark Field Scanning Transmission Electron Microscopy (HAADF-STEM) confirmed the spatial distribution of various elements within the nanocomposite. The representative TEM EDS elemental mapping analysis depicted in Figure 3g demonstrates the uniform distribution of Pt, Co, V, C, N, and O elements throughout the structure [35], respectively. TEM image characterization of PtCoV/g-C_3_N_4_ is shown in Appendix A. The crystal structure of the materials was examined using X-ray diffraction (XRD) analysis, as represented in Figure 3h. The peaks of 39.76°, 46.24°, and 67.46° belong to the (111), (200), and (220) crystal faces of Pt (PDF#04-0802), respectively [36]. For PtCoVO/g-C_3_N_4_, the diffraction peak of g-C_3_N_4_ can be observed, which fully indicates that PtCoVO and g-C_3_N_4_ are formed between chemical bonds. In the XRD pattern, there were fewer diffraction peaks associated with Pt, which might be caused by the low content of Pt. Electron paramagnetic resonance (EPR) is a shortcut and highly sensitive technique for monitoring the presence of oxygen vacancies [37]. PtCoVO/g-C_3_N_4_ exhibited a greater EPR signal strength at g = 1.97 (g = hv/βH) than PtCo-C_3_N_4_ (Figure 3i). The superior value of g can be attributed to oxygen vacancies on the catalytic material surfaces [38], which indicates the existence of surface oxygen vacancies on PtCoVO/g-C_3_N_4_.

In order to investigate the alterations in the electronic structure and elemental valence of PtCoVO/g-C_3_N_4_ and PtCoV/g-C_3_N_4_, X-ray photoelectron spectroscopy (XPS) analysis was conducted. The findings are presented in Figure 4a–d. The full XPS spectrum indicates the presence of Co, Pt, V, O, C, and N elements in the PtCoVO/g-C_3_N_4_ and PtCoV/g-C_3_N_4_ samples, corroborating the results obtained from the EDS analysis [39]. Figure 4a illustrates the XPS analysis of PtCoVO/g-C_3_N_4_, revealing two prominent peaks at 71.4 eV and 74.88 eV, which correspond to metallic Pt 4f_7/2_ and 4f_5/2_, respectively. Additionally, the presence of other peaks at 71.9 eV and 75.3 eV in the PtCoVO/g-C_3_N_4_ spectrum confirms the existence of oxidized states of Pt [40]. Compared to PtCoV/g-C_3_N_4_, the binding energy of PtCoVO/g-C_3_N_4_ displays a minor negative shift, which is related to the charge density change in the core metal ions [41]. In the case of V 2p, the peaks observed at binding energies of 521.23 eV and 525.18 eV in the PtCoVO/g-C_3_N_4_ spectrum (Figure 4b) can be assigned to V 2p_1/2_ and V 2p_3/2_ peaks, while a peak at 517.48 eV corresponds to V 2p_3/2_ in PtCoVO/g-C_3_N_4_, respectively [42]. In Figure 4c, the Co 2p spectra of PtCoVO/g-C_3_N_4_ exhibit two prominent peaks at 781.7 eV and 796.8 eV, corresponding to Co 2p_3/2_ and Co 2p_1/2_, respectively. Additionally, two shakeup satellite peaks (labeled as Sat.) are observed in the Co 2p spectra of PtCoVO/g-C_3_N_4_, located at 786.9 eV and 803.1 eV, indicating characteristic peaks associated with Co^2+^ species [43]. It can be observed that there is a peak at 781.7 eV in PtCoVO/g-C_3_N_4_ that can be designated as a Co-N bond. To further explain the existence of this Co-N bond, we tested the N 1s XPS spectra of PtCoVO/g-C_3_N_4_ (Appendix A), which were divided into 399.6 and 400.7eV for g-C_3_N_4_, corresponding to Co-N and graphite N, respectively [44]. Figure 4d presents the high-resolution spectrum of O 1s. In the PtCoVO/g-C_3_N_4_ spectrum, the binding energy of the M-OH bond is measured to be 530.9 eV [45]. In comparison to PtCoV/g-C_3_N_4_, PtCoVO/g-C_3_N_4_ exhibits a minor shift in the O 1s peak, with a forward shift of 0.5 eV for one sample and a negative shift of 0.6 eV for another sample. This observation suggests that the variation in vanadium species in PtCoVO/g-C_3_N_4_ resulted in an increased number of active sites, potentially altering electron transfer processes and enhancing the catalytic performance of PtCoVO/g-C_3_N_4_.

Ultraviolet photoelectron spectroscopy (UPS) was employed to examine the energy band structures of the synthesized H_x_V_2_O_5_, PtCoVO/g-C_3_N_4_, and PtCoV/g-C_3_N_4_ samples [46]. As depicted in Figure 4e, the UPS results reveal a strong correlation between the doping of Pt and Co metals and the secondary electronic cutoff edge of the catalytic materials [47]. The work function (W_F_) of a semiconductor is the difference between the Fermi level (E_F_) and the energy level E_0_ of the electron at rest in a vacuum [48,49]. The W_F_ of a material is typically around half the ionization energy of its corresponding metal-free atom. The W_F_ can be computed using the formula provided in Note S1 [50]. The magnitude of the W_F_ reflects the strength of electron binding within the metal. A smaller W_F_ indicates that electrons are more likely to be released from the metal [51]. Pt possesses the highest W_F_ of 5.65 eV. This high work function is advantageous in demonstrating the ease with which catalytic materials can be prepared by electronic means, resulting in superior electrochemical catalytic performance. As calculated from the UPS results, the PtCoVO/g-C_3_N_4_ presents an E_F_ of 9.37 eV [52,53]. Compared with PtCo/g-C_3_N_4_ (W_F_ = 5.2 eV) and PtCoV/g-C_3_N_4_ (W_F_ = 4.83 eV), PtCoVO/g-C_3_N_4_ has the lowest W_F_ = 4.42 eV. By examining the slope of the VB-XPS curve displayed in Figure 4f, we can observe the valence band maximum (VBM), denoted as EVB, of different materials. Specifically, H_x_V_2_O_5_, PtCo/g-C_3_N_4_, and PtCoVO/g-C_3_N_4_ have EVB values of 1.48 eV, −0.48 eV, and −1.20 eV, respectively. These data suggest that PtCoVO/g-C_3_N_4_ exhibits the most negative EVB value among the three materials, resulting in a higher Schottky barrier at the metal/interface for charge transport. Appendix A display the UPS spectra of PtCoVO/g-C_3_N_4_ at various levels of g-C_3_N_4_ added, alongside the VB-XPS spectrum.

### 2.3. HER Electrocatalytic Performance Analyses

To assess the electrocatalytic characteristics of the prepared materials, the HER properties were examined using a conventional three-electrode system in a 1 M potassium hydroxide solution. This evaluation aimed to gain insights into the electrocatalytic performance of the polymetallic-doped materials. To facilitate comparison, a commercially available Pt/C 20% catalyst was utilized as a reference. To ensure consistency, all potentials were calibrated with respect to the reversible hydrogen electrode (RHE). This standardization allows for accurate and meaningful comparisons between the different catalysts. According to the polarization curves plotted in Figure 5a, Pt/C 20% has the lowest overpotential of η_10_ = 90.6 mV to drive a current density of 10 mA cm^−2^, as expected [54,55]; contrarily, the bimetallic material (PtCo/g-C_3_N_4_) demonstrates a higher overpotential, specifically η_10_ = 386 mV, which indicates poor activity in comparison. In our this work, all trimetallic catalysts performed better than bimetallic materials. The catalytic performance of PtCoVO/g-C_3_N_4_ was found to be exceptional, as evidenced by its significantly lower overpotential of η_10_ = 92 mV when compared to that of PtCoV/g-C_3_N_4_ (η_10_ = 346 mV). This remarkable improvement of 254 mV highlights the superior activity of PtCoVO/g-C_3_N_4_ as a catalyst.

However, this difference in activity between PtCoVO/g-C_3_N_4_ and PtCoV/g-C_3_N_4_ could be a result of the higher catalyst activity due to the insertion of H_x_V_2_O_5_ (Figure 5a). This result was further confirmed by the optimization of the PtCoVO/g-C_3_N_4_ element ratio as the HER activity exhibited an upward trend with increasing H_x_V_2_O_5_ content. Notably, the HER activity reached its maximum when the ratio of Pt:Co:V was 0.1:0.4:0.4 (Appendix A). In addition, we also explored the polarization curves of Pt prepared by adding vanadium powder in different proportions (Appendix A), which also confirmed that the overpotential is the lowest at a ratio from 0.1 to 0.4. Thus, the PtCo/g-C_3_N_4_ and PtCoV/g-C_3_N_4_ used for comparison were prepared in this ratio. The catalytic mechanism of the HER can also be revealed by Tafel plots [56,57]. In general, there are three classical reactions for H_2_ evolution: electrochemical hydrogen ion adsorption (Volmer reaction), chemical desorption (Tafel reaction), and electrochemical desorption (Heyrovsky reaction) [58,59]. The first step in the HER is always the Volmer reaction, but the next step is either the Heyrovsky reaction or the Tafel reaction [60]. The calculated Tafel slopes of all samples are given in Figure 5b. The PtCoVO/g-C_3_N_4_ material displays the smallest Tafel slope (65.21 mV dec^−1^) among all tested materials, excluding Pt/C 20%. This result confirms the faster kinetics of the PtCoVO/g-C_3_N_4_ catalyst, indicating its superior performance in facilitating the hydrogen evolution reaction. However, PtCoVO/g-C_3_N_4_ exhibits a lower Tafel slope, thus demonstrating a much faster and efficient process of the HER compared to other compositions [61]. The verification of this result is supported by the comparison of performance, Tafel slopes, and overpotentials of the PtCoVO/g-C_3_N_4_ catalyst with some previously reported electrocatalytic materials, as shown in Appendix A. This comparison further confirms the superior performance and efficiency of PtCoVO/g-C_3_N_4_ in catalyzing the desired electrochemical reactions. Zhang et al. reported the good synergy between NiCo_2_O_4_ and C_3_N_4_, as many electrons tend to transfer from the hydrophobic region (NiCo_2_O_4_) to the electrophilic region (C_3_N_4_) and then to water, resulting in electron accumulation in water. DFT calculations shows that the E_H2O*_ and ΔG_H*_ of NiCo_2_O_4_@C_3_N_4_ are decreased [62]. In the study of Lu, the formed C_3_N_4_ layer with a large degree of graphitization can not only improve the conductivity of the catalyst, but can also prevent the corrosion of the active tungsten dioxide nanorods during the HER, exerting a protective effect [63]. Zhao et al. reported that CuSCN/C_3_N_4_ exhibited an overpotential of 85 mV, which is much higher than that of pure CuSCN. An interfacial N-Cu-S coordination mode is formed, which can promote electron penetration from the CuSCN crystal into the N atoms of the C_3_N_4_ shell, thereby improving the HER reactivity [64]. This indicates that these materials have a good synergistic effect with C_3_N_4_ (Appendix A). By adjusting the surface structure of transition metal atoms, reasonable design can obtain low-cost and efficient electrocatalysts, which is the key to developing an electrocatalytic HER. 

The Nyquist diagram, depicted in Figure 5c, was obtained through electrochemical impedance spectroscopy (EIS) measurements. The measured data align closely with the equivalent circuit model comprising charge transfer resistance (Rct), solution resistance (Rs), and a constant phase element (CPE), as illustrated in the insert of Figure 5c. The EIS results revealed that the Rct of the PtCoVO/g-C_3_N_4_ catalyst was significantly lower in the lower frequency region compared to that of PtCo/g-C_3_N_4_ and PtCoV/g-C_3_N_4_. This indicates that PtCoVO/g-C_3_N_4_ exhibited the highest electronic conductivity for the HER process, facilitating the fastest charge transfer rate and displaying the most sensitive electrocatalytic kinetics. Therefore, the resistance obtained from the EIS spectra of various materials shows that the PtCoVO/g-C_3_N_4_ catalyst has obvious lower impedance, which explains that the PtCoVO/g-C_3_N_4_ catalyst has faster reaction kinetics than the PtCo/g-C_3_N_4_ and PtCoV/g-C_3_N_4_ electrodes. In addition, the EIS spectra of different H_x_V_2_O_5_ contents proved the minimum impedance at a millimolar ratio of 0.1:0.4:0.4, as shown in Appendix A. Moreover, the EIS measurements showed that PtCoVO/g-C_3_N_4_ possess the smallest electron transfer resistance (the charge transfer resistance decreases as the arc size decreases). The trend observed in EIS for solid solutions is generally consistent with the trend of HER activity [65,66].

The double-layer capacitance (C_dl_) of PtCo/g-C_3_N_4_, PtCoVO/g-C_3_N_4,_ and PtCoV/g-C_3_N_4_ was evaluated through cyclic voltammetry (CV) measurements performed at different scan rates ranging from 0.02 to 0.1 V [67]. Appendix A provide the electrochemical double-layer capacitance of various materials. Based on the resulting C_dl_ values depicted in Figure 5d, PtCoVO/g-C_3_N_4_ has the highest value of all samples; this finding further validates the enhanced performance of PtCoVO/g-C_3_N_4_. The electrochemically active surface area (ECSA) of PtCo/g-C_3_N_4_ (28.95 cm^2^), PtCoVO/g-C_3_N_4_ (81.05 cm^2^), and PtCoV/g-C_3_N_4_ (42.88 cm^2^) was estimated by the C_dl_.

The XPS spectra shown in Figure 6a–d provide clear evidence that the PtCoVO/g-C_3_N_4_ material maintains excellent stability after undergoing electrocatalysis, providing a possible solution to the problem of durability in industrial applications. This statement can also be proved by the information in Appendix A. The chronoamperometry curve for PtCoVO/g-C_3_N_4_ demonstrates that the current density of PtCoVO/g-C_3_N_4_ is maintained at a constant value, with a minimal decrease observed only after 48 h of continuous operation, indicating the exceptional long-term electrochemical stability of PtCoVO/g-C_3_N_4_ (Figure 6e) [68]. The TEM image, as shown in the inset of Figure 6e, confirms that the morphology of PtCoVO/g-C_3_N_4_ remained intact, with minimal changes. Furthermore, the long-term durability test of PtCoVO/g-C_3_N_4_ was evaluated by potential cycling in 1 M KOH. As shown in Figure 6f, no significant loss of activity occurred after 5000 cycles and the value remained overlapped with the previous data, highlighting the PtCoVO/g-C_3_N_4_ catalyst’s excellent electrocatalytic stability [69]. The stability of the HER electrocatalyst properties of PtCoVO/g-C_3_N_4_ can be attributed to the notable influence of oxygen vacancies in H_x_V_2_O_5_ on the electrical properties and optimized structural properties. 

## 3. Experimental Methods 

### 3.1. Materials

Cobalt acetylacetonate (C_15_H_21_CoO_6_, 98%), chloroplatinic acid hexahydrate (H_2_PtCl_6_·6H_2_O, 99%), oil amine (C_18_H_37_N, 90%), vanadium powder (V, 200 mesh, 99.999%), potassium hydroxide (KOH, 90%), absolute ethanol (EtOH, 99.7%), hydrogen peroxide (H_2_O_2_, 30%), and urea (CH_4_N_2_O, 99%) were used. The materials mentioned above were acquired from Adamas Reagent. The Nafion solution (5wt.%, containing 15–20% water) was purchased from Sigma-Aldrich (Burlington, MA, USA). The aforementioned materials were utilized directly without undergoing any purification. The deionized water employed in the experiments was obtained by passing it through an ultrapure purification device.

### 3.2. Methods 

#### 3.2.1. Fabrication Process for g-C_3_N_4_ Nanosheets

Bulk *g*-C_3_N_4_ was prepared by the thermal treatment of urea. In the typical fabrication process, 5.0 g of urea was ground into a powder and subjected to calcination at 550 °C for 3 h, and the heating rate during this process was maintained at 5 °C/min. After cooling, a yellow solid was obtained and ground into powder for 30 min until the yellow color of *g*-C_3_N_4_ changed into a light-yellow color. 

#### 3.2.2. Synthesis of H_x_V_2_O_5_ HMOs

Material Oxides Synthesis: A total of 0.5 g of vanadium powder was dispersed in 50 mL of ethanol; during the dispersion process, continuous stirring was maintained for several minutes. Next, the vanadium metal suspension solution was supplemented with 2.5 mL of H_2_O_2_ solution. Following a reaction period of 3 h, the vanadium oxide solution underwent a color change from orange to eventually brown. The solution was subsequently dried using a vacuum chamber and then ground for subsequent utilization.

#### 3.2.3. Fabrication of PtCo/g-C_3_N_4_, PtCoV/g-C_3_N_4,_ and PtCoVO/g-C_3_N_4_ Catalysts

(1)Preparation process for PtCo/g-C_3_N_4_ materials

In a conical flask, 10 mL of water, 5 mL of EtOH, and 2 mL of oil amine were added. Then, 40 mg of g-C_3_N_4_, 142.5 mg of cobalt acetylacetonate, and 51.79 mg of chloroplatinic acid hexahydrate were weighed and introduced into the aforementioned solution. After subjecting the mixture to ultrasonic treatment for 60 min, it was transferred to a 25 mL reactor and maintained at 160 degrees Celsius for 16 h. The resulting solution was then allowed to cool naturally. When the temperature dropped down to room temperature, the product was obtained by centrifugation, washing, and drying. (Note: This material is named PtCo/g-C_3_N_4_ not because there is no oxygen element in the synthetic material, but in contrast with PtCoVO/g-C_3_N_4_, the original material of this catalyst is vanadium powder, whereas the original material of PtCoVO/g-C_3_N_4_ is H_x_V_2_O_5_, so the name distinguishes it. The same is true for PtCoV/g-C_3_N_4_).

(2)Preparation of PtCoVO/g-C_3_N_4_ materials

The preparation method for this material is similar to that of PtCo/g-C_3_N_4_, with the addition of H_x_V_2_O_5_. 

(3)Preparation of PtCoV/g-C_3_N_4_ materials

The preparation method for this material is similar to that of PtCoVO/g-C_3_N_4_, with the difference being that vanadium powder is used instead of H_x_V_2_O_5_.

## 4. Conclusions

By controlling the reaction rate, hydrogen metal oxides bronzes (HMOs), namely, the hydrogen vanadium bronze (H_x_V_2_O_5_), were obtained. Its oxygen content has a significant effect on its electrical properties. Subsequently, a novel hybrid nanocomposite consisting of g-C_3_N_4_ doped with a multi-metal was synthesized using a solvothermal method. Compared with other catalysts, the obtained PtCoVO/g-C_3_N_4_ catalyst possesses the characteristics of close contact between non-metals and metals, remarkable activity, fast electron transport, easy release of bubbles generated, and exceptional electrocatalytic durability. This study of the design of high-efficiency catalysts for the HER provides a novel concept, detailing the theoretical analysis to guide the experiment and the experimental results to support the theoretical calculation, with the theory and experiment complementing each other. The present study introduces an efficient and robust catalyst for integrated catalysts, along with a scalable insertion preparation technique and design concept. These advancements open up possibilities for broader applications in the fields of electrocatalysis and material science.

## Figures and Tables

**Figure 1 molecules-29-01518-f001:**
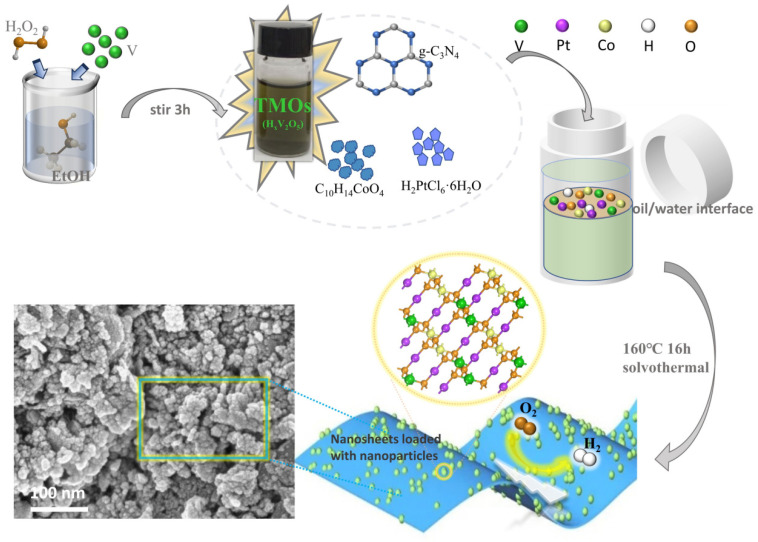
Schematic illustration of the synthesis process of PtCoVO/g-C_3_N_4_ nanocomposites at the oil/water interface.

**Figure 2 molecules-29-01518-f002:**
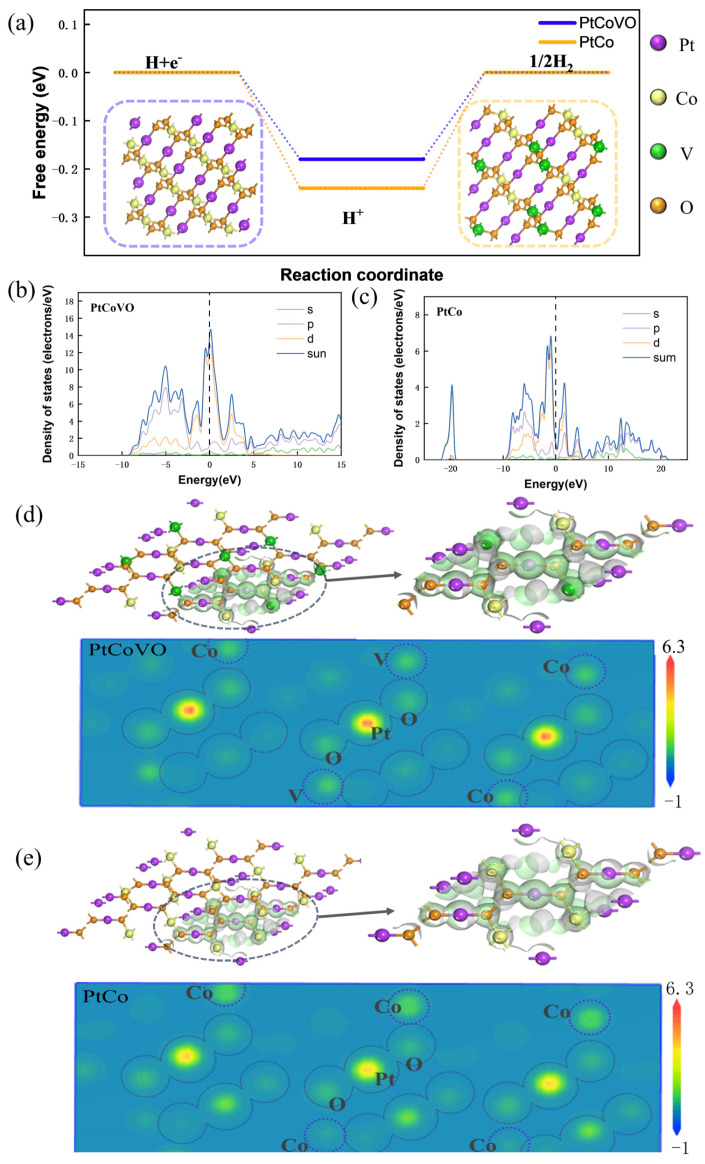
(**a**) Free energy diagrams for the HER of PtCo and PtCoVO. The diagram depicts the molecular structure of PtCo and PtCoVO. (**b**,**c**) The calculated DOS of PtCoVO and PtCo. (**d**,**e**) The charge density distribution of PtCoVO and PtCo.

**Figure 3 molecules-29-01518-f003:**
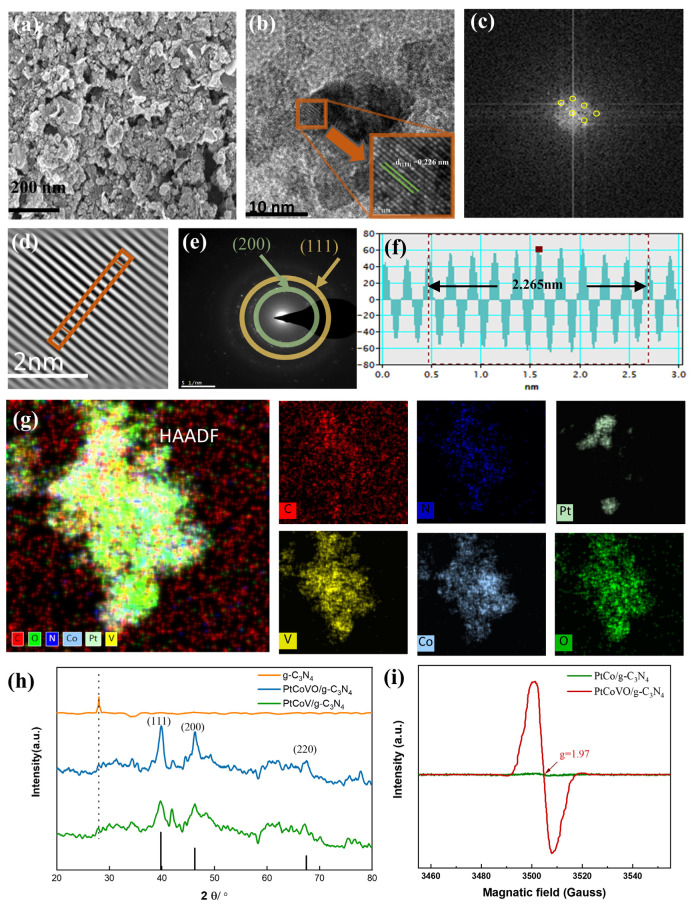
(**a**) SEM images of PtCoVO/g-C_3_N_4_. (**b**) TEM images of PtCoVO/g-C_3_N_4_. (**c**,**d**) FFT and IFFT pattern of PtCoVO/g-C_3_N_4_. (**e**) SAED images of PtCoVO/g-C_3_N_4_. (**f**) The PtCoVO/g-C_3_N_4_ spacing was 0.2265 nm. (**g**) HAADF of PtCoVO/g-C_3_N_4_ and elemental mapping of C, N, Pt, V, Co, and O, respectively. (**h**) XRD of PtCoVO/g-C_3_N_4_ and PtCo/g-C_3_N_4_, standard card (PDF#04-0802). (**i**) EPR spectra of PtCo/g-C_3_N_4_ and PtCoVO/g-C_3_N_4_ at 103 K in liquid N_2_.

**Figure 4 molecules-29-01518-f004:**
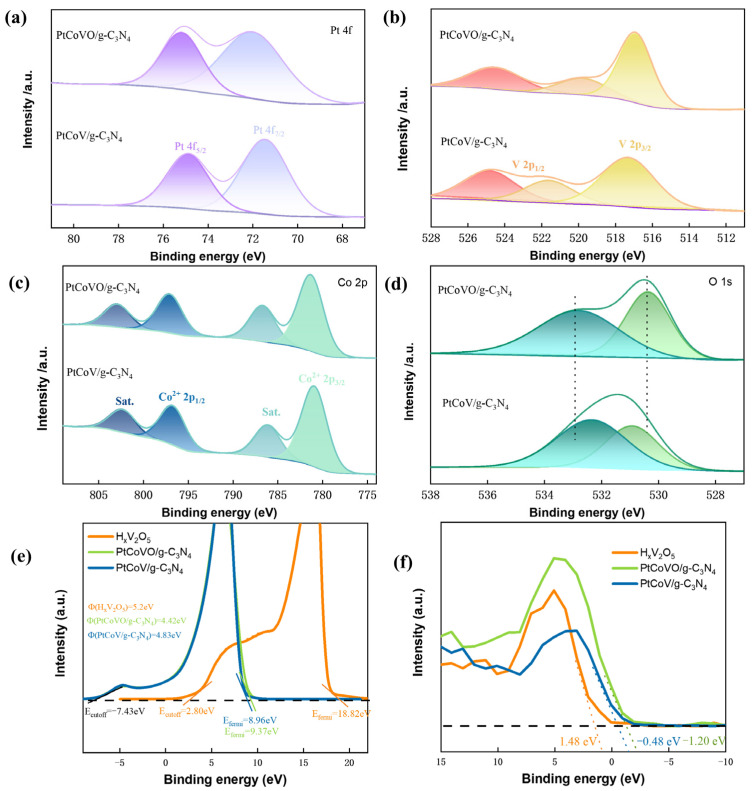
XPS spectra of PtCoVO/g-C_3_N_4_ and PtCoV/g-C_3_N_4_. (**a**) Pt 4f. (**b**) V 2p. (**c**) Co 2p. (**d**) O 1s. (**e**) UPS spectra of H_x_V_2_O_5_, PtCoVO/g-C_3_N_4,_ and PtCoV/g-C_3_N_4_. (**f**) VB-XPS valence band spectra of H_x_V_2_O_5_, PtCoVO/g-C_3_N_4,_ and PtCoV/g-C_3_N_4_.

**Figure 5 molecules-29-01518-f005:**
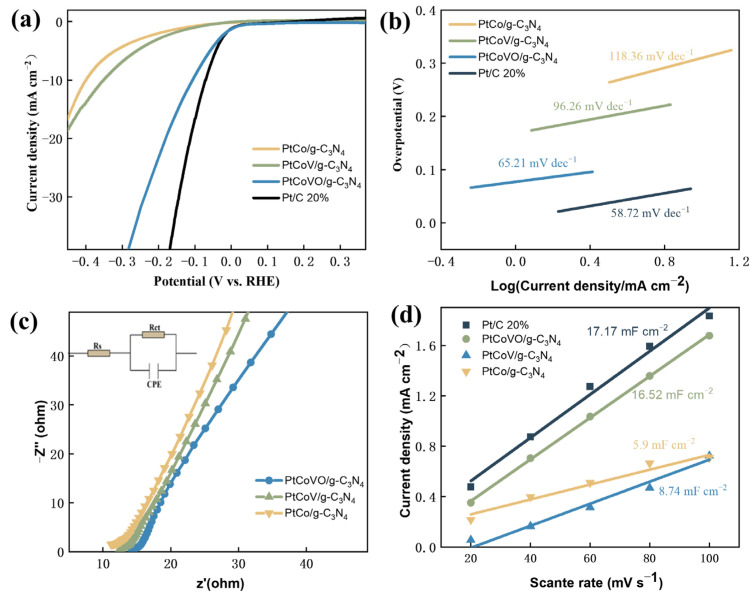
(**a**) HER polarization curves. (**b**) Tafel plots of PtCo/g-C_3_N_4_, Pt/C 20%, PtCoVO/g-C_3_N_4_, and PtCoV/g-C_3_N_4_. (**c**) EIS Nyquist plots of PtCoVO/g-C_3_N_4_, PtCo/g-C_3_N_4_, and PtCoV/g-C_3_N_4_. (**d**) C_dl_ values of current density differences plotted against scan rates of Pt/C 20%, PtCoVO/g-C_3_N_4_, PtCo/g-C_3_N_4,_ and PtCoV/g-C_3_N_4_.

**Figure 6 molecules-29-01518-f006:**
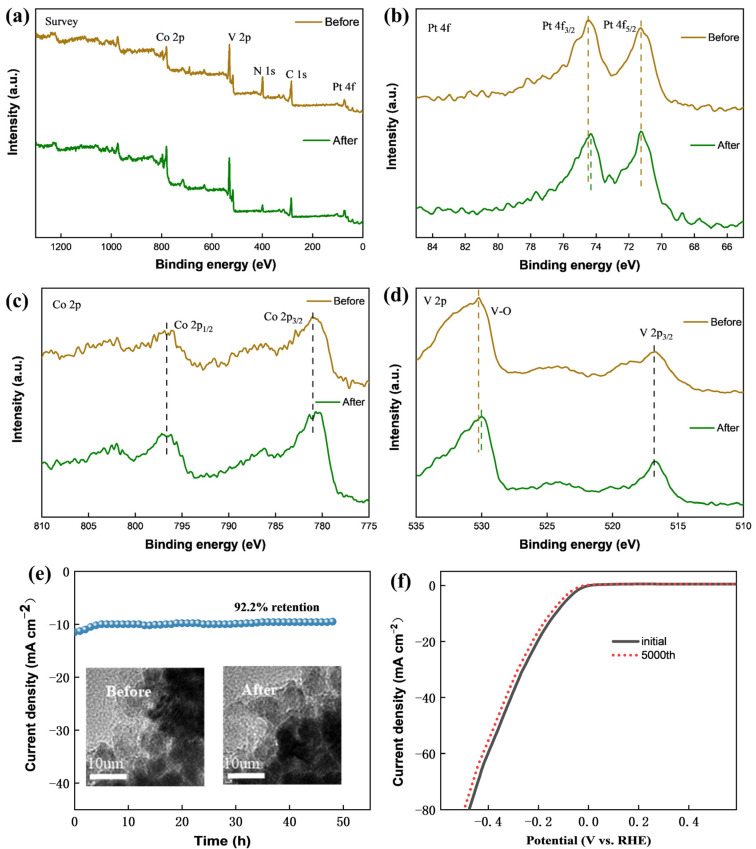
(**a**–**d**) XPS spectra of survey, Pt 4f, Co 2p, and V 2p for PtCoVO/g-C_3_N_4_ before and after electrochemical testing. (**e**) The PtCoVO/g-C_3_N_4_ catalyst was subjected to a long-term stability measurement at current densities of 10 mA cm^−2^ for 48 h; the inset displays the TEM images before and after the stability measurement. (**f**) Continuous CV scanning after 5000 circles at 100 mV s^−1^ of PtCoVO/g-C_3_N_4_.

## Data Availability

Data are contained within the article and the Appendix A.

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
