# Peer review of "Intercalated PtCo Electrocatalyst of Vanadium Metal Oxide Increases Charge Density to Facilitate Hydrogen Evolution"

_molecules, 2024, doi:10.3390/molecules29071518_

Round 1

Reviewer 1 Report

Comments and Suggestions for Authors

The work is interesting and describes a topic related to electrocatalysis, specifically focusing on the performance of an electrocatalyst for the hydrogen evolution reaction. Hydrogen energy is receiving increasing attention, making the work highly relevant. 

Some corrections are needed. 

line 18 - the description of the abbreviation (g-C3N4) is needed (never mentioned from the beginning till the end of the manuscript).

line 38 - Ref. 7 neither relevant as oxides nor alloys.

Ref. 10. 11 are review articles, not directly research articles on nitride-based HER catalysts.

line 56 - Previous reports have shown.... which previous reports, please cite

line 64 - please provide information about the impact of present oxygen vacancies in HVOs by referencing.

line 84 - Ref. 19 does not seem relevant to this synthesis process. How ref. 19 is related?

line 108, 113 - why Ref. 21,22 are cited? Novelty of using Ref 21. How are Ru and N-Ru related to Pt and PtCo?

line 115, 136 - the description of the abbreviation (HMOs, PDOS) is needed.

line 271-273 - different size font

line 287 - "in the study of Lu" it would be better to say Lu et al., because the cited paper has more than one author.

Cdl - dl should be in subscript

line 384- "Then add 2.5 ml in the vanadium solution of hydrogen peroxide solution. Next, the vanadium metal suspension solution was supplemented with 2.5 ml of H2O2 solution." So how much H2O2 was used?

line 390,403 - C3N4, 3 and 4 should be in subscript.

subscript is needed lines: 449 (C76), 455 (MoS2 and WS2); 474  (M-1); and 477 appropriately written (Pt-Ox)-(Co-Oy).

Could you please give me an explanation of how the Tafel slope was calculated? What potential ranges were used to obtain the presented data? If I got right, according to your given results the overpotentials different at 10 mAcm-2 between PtCoVO/g-C3N4 and Pt/C 20% is less than 2 mV. 

Reviewer 2 Report

Comments and Suggestions for Authors

In this manuscript, the authors reported a universal strategy for the rapid synthesis of a highly active hydrogen evolution reaction (HER) electrocatalyst using a controllable hydrogen insertion method and solvothermal process. Hydrogen vanadium bronze HxV2O5 was first obtained through controlling the ethanol reaction rate in the oxidization process of hydrogen peroxide, followed by a solvothermal method at the oil/water interface that supported the intermetallic PtCoVO on two-dimensional g-C3N4 nanosheets (PtCoVO/g-C3N4). The as obtained PtCoVO/g-C3N4 demonstrated superior HER characteristics compared to PtCo/g-C3N4 and PtCoV/g-C3N4. Insights into the improved HER performance was further obtained through combined experimental and theoretical studies. Overall, this work has very good novelty and the results were thoroughly presented. I believe this manuscript is suitable for the journal Molecules. However, to further improve the quality and clarity of the manuscript, the below detailed comments need to be addressed.

1. Please clarify the nature of the catalyst. Is it a metal alloy or metal oxide? Is it partially oxidized alloy? While most of the evidence and discussion suggest that it is a nanoalloy, the labelling of PtCoVO might make people confused. The authors are suggested to properly define the material.

2. To appeal to a broader readership, recent works on water electrolysis (Small Methods, 2022, 6, 2201099; Chem. Eng. J., 2023, 471, 144660) are recommended to be included in the Introduction.

3. Figure 5b, there is an obvious inconsistency in the data presented here compared with that presented in Figure 5a. (1) Why would PtCoV/g-C3N4 show the best performance in Figure 5b? The activity trend was not consistent with that shown in Figure 5a. (2) If one compares the overpotential at 10 mA cm-2 (i.e., log i = 1), one will find the overpotential for all samples in Figure 5b does not correspond to that in Figure 5a. Please carefully double check Figure 5b and make revision accordingly.

4. Line 273, the authors described “their OER curves have been observed in Figure S24”. Why was OER studied? This seems irrelevant to the current work since it is about HER. Then, Figure S24 is also not relevant. Please reconsider this part.

5. Figure 3h, what material does the standard peak refer to? This should be mentioned either in the figure or in the figure caption. What about the XRD of the g-C3N4? The authors might wish to include for comparison. Also, Figure 3h and 3i, the samples were not labelled the same way as in other figures (the “g” was missing for “g-C3N4”).

6. The references are insufficient. Recent works on HER can be included (e.g., Materials Reports: Energy, 2022, 2, 100144).

7. Figure 4a, please double check the labelling of the Pt 4f7/2 and Pt 4f5/2, which might have been swapped. Also, please note that the peak area ratio should follow Pt 4f7/2 : Pt 4f5/2 = 4:3 when fitting the peaks.

8. The format of several figures need to be revised. (1) Figure S1 shows the digital image of vanadium powder in hydrogen peroxide ethanol solution at hourly intervals, but the exact time for each sample was not labelled. (2) For figures S4, 7, 8, 13, the unit for the x axis (Band energy) was not given. Please double check. (3) Figure 4f, the legend wrote “HV2O5”, which appeared to be a typo and should be revised into “HxV2O5”. (4) Figure 2 d, e, what is the unit for the scale shown on the right side of the figure (from -1 to 6.3)? (5) Figure 1, the scale bar for the electron microscopy image should be provided.

9. More experimental details should be provided. For instance, how was the potential vs Ag/AgCl converted to that against the RHE?

10. A bit more keywords can be included to increase the visibility of the manuscript.

Author Response

Thank you for your great help! We feel very grateful.

We are very thankful for the comments from both the editor and reviewers. Based on these comments and suggestions, careful revisions have been made on the original manuscript, in the hope that the issues raised may be duly addressed. The responses to reviews and detailed changes are listed in response letters.

Round 2

Reviewer 1 Report

Comments and Suggestions for Authors

The explanation for the fig. 5 it's not clear for me and maybe it gives even more doubts now. 

Authors replay: "The measured potentials vs SCE were turned into vs.

reversible hydrogen electrode (RHE) according the equation: ERHE = EAg/AgCl +0.0592pH+ 0.197. Plot the LSV curve as the logarithm of the standard potential and current density to obtain the Tafel diagram and the Tafel is calculated by the equation:

η = a + b × log j Where η represents the overpotential, a represents the Tafel constant, b represents the Tafel slope, and j represents current density. 

Why in equation potential is given vs. Ag/AgCl, when the measured potential is given SCE. The conversion of potentials from SCE to RHE using the equation ERHE = EAg/AgCl + 0.0592pH + 0.197 should be reviewed and checked for accuracy. If there's any doubt or confusion, it's essential to verify the correctness of this conversion.

Tafel slope equation is explained, but my question is how you calculated it, what potential rage was taken for Tafel plots? 

The Tafel plots now looks different than in previous version, but the Tafel slope values haven't changed, is it correct or it's mistake?

Typically, to calculate the Tafel slope, the potential range is recommended to be at least 100 mV, with an ideal range falling within 100-300 mV. 

In authors case, it's obvious that potential range is less than 100mV. 

It's advisable for the authors to review and revise their calculations, ensuring accuracy in the conversion of potentials and the determination of Tafel slopes. 

Author Response

Thank you for your great help! We feel very grateful.

We are very thankful for the comments from both the editor and reviewers. Based on these comments and suggestions, careful revisions have been made on the original manuscript, in the hope that the issues raised may be duly addressed. The responses to reviews and detailed changes are listed in attachment.
